# $H_\infty$-based control of multi-agent systems: Time-delayed signals, unknown leader states and switching graph topologies

**Amin Taghieh** [1], **Ardashir Mohammadzadeh** [2]*, **Sami ud Din** [3], **Saleh Mobayen** [4]*, **Wudhichai Assawinchaichote** [5]*, **Afef Fekih** [6]

**1** Department of Electrical Engineering, Qatar University, Doha, Qatar, **2** Electrical Engineering Department, University of Bonab, Bonab, Iran, **3** Department of Electrical Engineering, Namal University Mianwali, Mianwali, Pakistan, **4** Future Technology Research Center, National Yunlin University of Science and Technology, Douliu, Taiwan, **5** Department of Electronic and Telecommunication Engineering, Faculty of Engineering, King Mongkut's University of Technology Thonburi, Bangkok, Thailand, **6** Department of Electrical and Computer Engineering, University of Louisiana at Lafayette, Lafayette, Louisiana, United States of America

* a.mzadeh@ubonab.ac.ir, intelligent.controlref@gmail.com (AM); mobayens@yuntech.edu.tw (SM); wudhichai.asa@kmutt.ac.th (WA)

**Data Availability Statement:** All relevant data are within the manuscript.

## Abstract

The paper investigates a leader-following scheme for nonlinear multi-agent systems (MASs). The network of agents involves time-delay, unknown leader's states, external perturbations, and switching graph topologies. Two distributed protocols including a consensus protocol and an observer are utilized to reconstruct the unavailable states of the leader in a network of agents. The $H_\infty$-based stability conditions for estimation and consensus problems are obtained in the framework of linear-matrix inequalities (LMIs) and the Lyapunov-Krasovskii approach. It is ensured that each agent achieves the leader-following agreement asymptotically. Moreover, the robustness of the control policy concerning a gain perturbation is guaranteed. Simulation results are performed to assess the suggested schemes. It is shown that the suggested approach gives a remarkable accuracy in the consensus problem and leader's states estimation in the presence of time-varying gain perturbations, time-delay, switching topology and disturbances. The $H_\infty$ and LMIs conditions are well satisfied and the error trajectories are well converged to the origin.

## Introduction

Inspired by the energy-minimization strategy during bird migration, leader-following (LF) consensus or coordinated tracking problem has been a topic of miscellaneous research spheres in multi-agent systems (MASs) [1–3]. For instance, distributed tracking controllers have been applied to the networked Euler–Lagrange systems with a leader [4, 5]. Recently, estimating the attitude of each mobile robot via an observer and employing the leader-following strategy, trajectory tracking of mobile robots has been analyzed [6]. Moreover, the consensus problem of linear MASs under multiple targets/leaders has been investigated in [7]. Based on a sliding

**Funding:** The funders had no role in study design, data collection and analysis, decision to publish, or preparation of the manuscript.

**Competing interests:** The authors have declared that no competing interests exist.

mode strategy (SMC), an adaptive distributed scheme has been designed in [8] for the LF consensus problem. The consensus tracking problem of stochastic nonlinear MASs has been investigated by utilizing event-triggered mechanisms in [9]. Recently, the LF consensus of MASs with semi-Markov jump parameters has been analyzed by implementing a hybrid event-triggered strategy to tackle the transmission frequency of surplus data packets [10]. The LF problem of stochastic MASs subject to multiplicative noises has been studied via output feedback control policies in [11]. The consensus tracking problem of MASs in the presence of unknown dynamics/nonparametric uncertainties has been studied through designing a distributed control law in [12]. Based on an impulsive model, the fixed-time tracking control problem has been studied for a set of planar agents in a surveillance network in [13].

Although fixed network topologies are primarily considered in research papers, packet losses, channel fading, and data congestions may not be pragmatically fulfilled under this restriction [14, 15]. Concerning this, time-varying switched network topologies with a finite set of configurations are more realistic and demanding. Therefore, the consensus problem of linear time-varying and time-invariant MASs under connected communication graph and switching topologies (STs) have been studied [16–18]. A distributed adaptive protocol has been suggested for the LF consensus issue of linear time-varying MASs under STs [19]. Furthermore, the consensus of Lipschitz-type nonlinear MASs in second-order dimensions under STs has been studied in the literature [20, 21]. Distributed control policies have been proposed for mobile autonomous agents with leaders under switching directed network topologies in [22]. Considering the static positions for leaders and an undirected switching graph topology in [23], the convergence problem of followers to the convex hull has been studied. Based on a distributed control scheme, the LF consensus of MASs with switching topologies and stochastic disturbances has been analyzed in [24]. Utilizing an average dwell time condition and distributed control policies, the LF consensus problem of MASs with unknown control/output directions and switching topologies has been studied in [25]. For MASs with one-sided Lipschitz nonlinear dynamics, the LF consensus problem has been studied under switching topologies in [26]. Moreover, the LF consensus problem for MASs has been analyzed by designing an event-triggered control scheme in [27].

While in most of the above-mentioned contributions the leader's states are accessible, in actual operations, it is of utmost importance to fabricate a distributed state-estimation (DSE) to approximate the leader states. By proposing an estimating strategy for leaders in [28], a cooperative regulation scheme is scrutinized for linear MASs. An observer in the adaptive scheme has been proposed in [29] to estimate both the system's matrices and the leader's states. Although DSE mechanisms have been evolved for the target tracking problem of MASs, there are still remarkable open issues including the LF consensus problem of nonlinear MASs under STs.

On the other hand, the analysis becomes more complex if time-delay is involved in leader states and followers. Since the delay phenomenon strikes the system's performance and engenders instability, an appropriate DSE and distributed controller should be applied to the MASs. Lyapunov-Krasovskii functional (LKF) as a useful tool is utilized to investigate the stability of time-delayed systems in [30–34]. LF consensus problem of time-delay double integrator systems under switching interconnection graphs has been investigated in [35]. Designing a distributed observer, the cooperative containment control of linear MASs with time-delay has been studied in [36]. Moreover, a control protocol in a distributed scheme is presented in [37] to study chaotic MASs subject to time-delay.

Based on the above arguments, this paper researches a novel distributed state observer (DSO) and a distributed controller for nonlinear MASs under STs based on the consensus strategies. Since transmission time-delay exists in the states of the leader and the followers, an

appropriate LKF is employed. Simulations are accomplished to sketch the usefulness of the designed approaches. The significant contributions are listed as:

- In comparison with the LF consensus problem analysis or tracking problem investigation where the leader's or target's states are accessible, in this paper this assumption is violated and they are completely inaccessible; hence, an observer in distributed form is designed to reconstruct the leader states. Moreover, the network of the agents is subjected to time-delay.

- The switching topologies are considered for the communication network. In addition, the influence of delay in the states of the leader and each follower is investigated in this paper. Furthermore, due to the actuator degradations, time-varying gain perturbation is considered and robustness of the distributed controller for gain perturbations is studied.

- $H_\infty$ LF consensus problem of the time-delay nonlinear MAS with unknown leader's states is investigated based on a prescribed $H_\infty$ disturbance attenuation along with a DSO and a distributed controller.

## Preliminaries and problem definition

### Graph theory

A set of switching graphs $\sigma_\iota$ with alteration which is adjusted by a switching signal $\vartheta(t) \to \iota \in \{1, 2, \ldots, \ell\}$, represents interactions among a network of agents.

Let $\sigma_{\vartheta(t)} (B_{\vartheta(t)}, \gamma, A_{\vartheta(t)})$ denotes the interaction network; $\gamma = \{0, 1, \ldots, N\}$ represents the node set, $B_{\vartheta(t)} \subseteq \gamma \times \gamma \{(i, j) \in \gamma \times \gamma\}$ is the set of edges (where the pair of $(i, j) \in B_{\vartheta(t)}$ if the interconnection between nodes $i$ and $j$ exists, where $j$-th node is the neighbour of $i$-th node, or in other words $i$-th agent is able to attain information from $j$-th node, else $(i, j) \notin B_{\vartheta(t)}$), and $A_{\vartheta(t)} = [a_{\vartheta(t),ij}] \in R^{N \times N}$ denotes adjacency matrix. For element $a_{\vartheta(t),ij}$ of adjacency matrix $A_{\vartheta(t)}$, it is elucidated that $a_{\vartheta(t),ii} = 0$, $a_{\vartheta(t),ij} > 0$ if $(i, j) \in B_{\vartheta(t)}$, and $a_{\vartheta(t),ij} = 0$ otherwise. Furthermore, $L_{\vartheta(t)} = [l_{\vartheta(t),ij}] \in R^{N \times N} = D_{\vartheta(t)} - A_{\vartheta(t)}$ is the Laplacian matrix, where $D = \text{diag}\{d_{\vartheta(t)}^1, d_{\vartheta(t)}^2, \ldots, d_{\vartheta(t)}^N\}$ and $d_{\vartheta(t)}^i = \sum_{j=1}^N a_{\vartheta(t),ij}$. The diagonal matrix $\Xi_{\vartheta(t)} = \text{diag}\{a_{\vartheta(t),10}, a_{\vartheta(t),20}, \ldots, a_{\vartheta(t),N0}\}$ denotes the interconnection between the leader and the follower. If the $i$-th node receive data from the leader then $a_{\vartheta(t),i0} > 0$ otherwise $a_{\vartheta(t),i0} = 0$.

**Assumption 1**. [38]. *Consider graph $\sigma_\iota(\iota = 1, 2, \ldots, \ell)$, the connections among in-neighboring followers are assumed to be undirected.*

**Assumption 2**. [19]. *It is assumed that a directed spanning tree exists in $\bar{\sigma} = \bigcup_{i=1}^\ell \sigma_\iota$, rooted at leader.*

### Problem definition

Assume a network of time-delay agents with a LF framework within agents. The dynamics are written as:

$$\dot{x}_i(t) = Ax_i(t) + A_\tau x_i(t - \tau) + Ef(x_i(t)) + Bu_i(t) + Wd_i(t), \quad i = 1, \ldots, N \qquad (1)$$

where $x_i(t)/u_i(t)$ denote the state/controller of the agent $i$, respectively. $\tau$ stands for a given time-delay, the external disturbance $d_i(t)$ materializes in $l_2([0, +\infty); \mathbb{R}^q)$, and $f(.)$ satisfies the Lipschitz condition:

$$\|f(v) - f(\omega)\| \leq \gamma \|(v - \omega)\| \qquad (2)$$

where $\gamma$ is the Lipschitz constant. The leader's dynamic is presumed to be as

$$\dot{x}_0(t) = Ax_0(t) + A_\tau x_0(t - \tau) + Ef(x_0(t)) + A_n r(t) \tag{3}$$

where $x_0(t) \in \mathbb{R}^n$ is the leader's state. The measurements are given as:

$$y_i(t) = Cx_i(t) + C_n v_i(t) \tag{4}$$

where $r(t)$ and $v_i(t)$ are white Gaussian noises.

**Assumption 3**. *The pair $(A, B)/(A, C)$ is considered to be stabilizable/observable.*

**Lemma 1**. [39]. *For any scalar $\epsilon > 0$ and vectors $\mathcal{O}, \mathcal{T} \in \mathbb{R}^n$, one has*

$$2\mathcal{O}^T\mathcal{T} \leq \epsilon\mathcal{O}^T\mathcal{O} + \epsilon^{-1}\mathcal{T}^T\mathcal{T} \tag{5}$$

**Lemma 2**. [40]. *For any scalar $\alpha > 0$ and real matrices $\bar{\sigma}, \bar{H}, \bar{J}$ with appropriate dimensions and $\bar{J}^T\bar{J} < \varrho I$, the following inequality holds*

$$\bar{\sigma}\bar{J}\bar{H} + \bar{H}^T\bar{J}^T\bar{\sigma}^T \leq \alpha^{-1}\bar{\sigma}\bar{\sigma}^T + \alpha\varrho\bar{H}^T\bar{H} \tag{6}$$

The general scheme of deigned controller is depicted in Fig 1. The details are illustrated in following sections.

**Remark 1**. In two Theorems, the asymptotic stability is proved. It is shown that by the designed scheme, the followers asymptotically observe the leader, and the $H_\infty$-based leader following consensus is satisfied.

## Main results

The principal strategy is to formulate the observer/controller in distributed scheme for the MASs (1) to follow the state estimations of the leader (3). The suggested observer is written

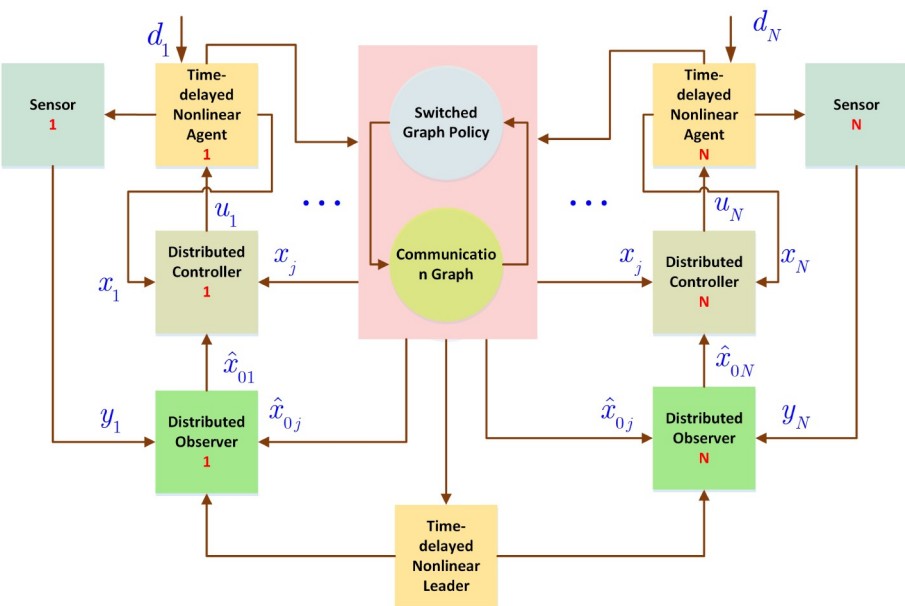

**Fig 1. General diagram of controller.**

as:

$$\dot{\hat{x}}_{0i}(t) = A\hat{x}_{0i}(t) + A_\tau\hat{x}_{0i}(t-\tau) + Ef(\hat{x}_{0i}(t)) - \Theta(y_i(t) - C\hat{x}_{0i})$$
$$+ BK\sum_{j\in N_i} a_{\vartheta(t),ij}(\hat{x}_{0i}(t) - \hat{x}_{0j}(t)), \quad i = 1,\ldots,N \tag{7}$$

where $\Theta \in \mathbb{R}^{n\times p}$ denotes the gain matrix of observer which should be computed, $\hat{x}_{0i}(t) \in \mathbb{R}^n$ represents the estimation of $x_0(t)$, $K \in \mathbb{R}^{n\times m}$ is a matrix which is employed to achieve stability conditions of suggested DSO scheme. The term $(y_i(t) - C\hat{x}_{0i})$ is utilized to the estimator updating and represents the function of measurement which has the new data. Moreover, the term $\sum_{j\in N_i} a_{\vartheta(t),ij}(\hat{x}_{0i}(t) - \hat{x}_{0j}(t))$ is utilized to employ the data of the neighbouring nodes for the state estimations (see Fig 2).

## Leader-following consensus

The following LF consensus protocol is implemented:

$$u_i(t) = \varphi(t)\left[\sum_{j\in N_i} a_{\vartheta(t),ij}(x_i(t) - x_j(t)) + a_{\vartheta(t),i0}(x_i(t) - \hat{x}_{0i}(t))\right]$$
$$, i = 1,\ldots,N \tag{8}$$

where $\varphi(t) = \varphi + \Delta\varphi(t)$, in which $\varphi$ is the $n \times m$ feedback gain matrix which should be computed and $\Delta\varphi$ is the controller gain perturbation which exists due to the sensing faults, actuator degradations, and roundoff errors and may cause performance deterioration or even system

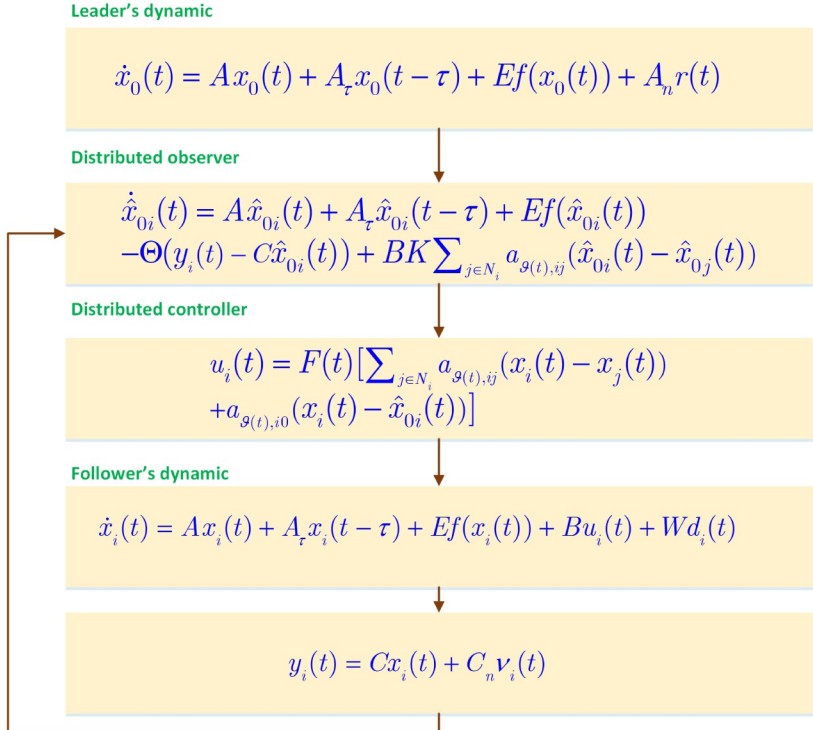

**Fig 2. Proposed observer-based scheme.**

destabilization. In this paper, the term $\Delta\varphi$ takes the following multiplicative form to endure norm-bounded uncertainties and allocate safe tuning margins simultaneous with synthesizing of the consensus protocol (8).

$$\Delta\varphi = M\Delta(t)N, \quad \Delta^T(t)\Delta(t) \le I \tag{9}$$

in which $\Delta(t)$ denotes the unknown time-varying matrix whilst $M$ and $N$ of appropriate dimensions represent known matrices.

**Theorem 1**. *Consider network of agents* (1) *with time-delay and* $d_i(t) = 0$ *and* (3) *utilizing the DSO* (7) *and the control policy* (8) *under switching graphs* $\sigma_{\vartheta(t)}$. *For given positive scalars* $\epsilon_1 : \epsilon_4$, $\alpha_1 : \alpha_3$, *suppose that there exist symmetric matrices* $P > 0, Q > 0, R > 0, S > 0$, *and the matrix X with suitable dimensions such that*:

$$
\begin{bmatrix}
\Omega_{1,1} & C^T X^T & PA_\tau & 0 & \sqrt{\psi_1}PB & \sqrt{\epsilon_1}PE & \sqrt{\psi_2}PBM & 0 & 0 \\
* & \Omega_{2,2} & 0 & QA_\tau & 0 & 0 & 0 & \sqrt{2\lambda}QB & \sqrt{2\lambda}QE \\
* & * & -R & 0 & 0 & 0 & 0 & 0 & 0 \\
* & * & * & -S & 0 & 0 & 0 & 0 & 0 \\
* & * & * & * & -I & 0 & 0 & 0 & 0 \\
* & * & * & * & * & -I & 0 & 0 & 0 \\
* & * & * & * & * & * & -I & 0 & 0 \\
* & * & * & * & * & * & * & -I & 0 \\
* & * & * & * & * & * & * & * & -I
\end{bmatrix} < 0
\tag{10}
$$

$$\Omega_{1,1} = A^T P + PA + \epsilon_1^{-1}\gamma^2 I + R + (\alpha_2\lambda^2 + (1 + \epsilon_4)\alpha_1\mu^2)N^T N$$
$$\Omega_{2,2} = A^T Q + QA + XC + C^T X^T + (\epsilon_2^{-1}\gamma^2 + \epsilon_3^{-1} + \epsilon_4^{-1})I + S$$
$$\psi_1 = 2\lambda + (1 + \epsilon_3)(\alpha_3^{-1} + \alpha_3\mu^2), \quad \psi_2 = \alpha_2^{-1} + (1 + \epsilon_4)\alpha_1^{-1}$$

*where* $\lambda = \max\{\lambda_i(L_\iota)\}$ *and* $\mu = \max\{a_{\iota,i0}\}$.

*Then, the followers asymptotically observe the leader* (3) *and the NMAS achieves the LF consensus.*

*Moreover the feedback gain of the control policy* (8) *is achieved as* $\varphi = B^T P$ *and the DSO gains in* (7) *are computed via* $K = B^T Q$ *and* $\Theta = Q^{-1} X$.

*Proof.* By substituting (8) into (1), we obtain:

$$
\begin{aligned}
\dot{x}_i(t) = {}& Ax_i(t) + A_\tau x_i(t - \tau) + Ef(x_i(t)) \\
& + B(\varphi + \Delta\varphi(t))\sum_{j\in N_i} a_{\vartheta(t),ij}(x_i(t) - x_j(t)) + B(\varphi + \Delta\varphi(t))a_{\vartheta(t),i0}(x_i(t) - \hat{x}_{0i}(t))
\end{aligned}
\tag{11}
$$

let $\upsilon_i(t) = x_i(t) - x_0(t)$ and $\zeta_i(t) = x_0(t) - \hat{x}_{0i}(t)$ be the consensus and estimation errors,

respectively. Regarding (3), (7), and (11) one can achieve that

$$
\begin{aligned}
\dot{v}_i(t) &= Av_i(t) + A_\tau v_i(t-\tau) + E(f(x_i(t)) - f(x_0(t))) \\
&\quad + B(\varphi + \Delta\varphi(t))\sum_{j\in N_i} a_{\vartheta(t),ij}(v_i(t) - v_j(t)) + B(\varphi + \Delta\varphi(t))a_{\vartheta(t),i0}(v_i(t) + \zeta_i(t)) \\
\dot{\zeta}_i(t) &= A\zeta_i(t) + A_\tau \zeta_i(t-\tau) + E(f(x_0(t)) - f(\hat{x}_{0i}(t))) + \Theta C(v_i(t) + \zeta_i(t)) \\
&\quad + BK\sum_{j\in N_i} a_{\vartheta(t),ij}(\zeta_i(t) - \zeta_j(t))
\end{aligned}
\tag{12}
$$

where

$$
\Phi(x(t)) =
\begin{bmatrix}
(f(x_1(t)) - f(x_0(t)))^T \\
\vdots \\
(f(x_N(t)) - f(x_0(t)))^T
\end{bmatrix}
\tag{13}
$$

$$
\Psi(x(t)) =
\begin{bmatrix}
(f(x_0(t)) - f(\hat{x}_{01}(t)))^T \\
\vdots \\
(f(x_0(t)) - f(\hat{x}_{0N}(t)))^T
\end{bmatrix}
\tag{14}
$$

According to Kronecker product, the compact form of (12) is acquired as

$$
\begin{aligned}
\dot{v}(t) &= (I_N \otimes A)v(t) + (I_N \otimes A_\tau)v(t-\tau) + (I_N \otimes E)\Phi(x(t)) \\
&\quad + (L_{\vartheta(t)} \otimes B(\varphi + \Delta\varphi(t)))v(t) + (\Xi_{\vartheta(t)} \otimes B(\varphi + \Delta\varphi(t)))v(t) \\
&\quad + (\Xi_{\vartheta(t)} \otimes B(\varphi + \Delta\varphi(t)))\zeta(t) \\
\dot{\zeta}(t) &= (I_N \otimes A)\zeta(t) + (I_N \otimes A_\tau)\zeta(t-\tau) + (I_N \otimes E)\Psi(x(t)) \\
&\quad + (I_N \otimes \Theta C)v(t) + (I_N \otimes \Theta C)\zeta(t) + (L_{\vartheta(t)} \otimes BK)\zeta(t)
\end{aligned}
\tag{15}
$$

Regarding Assumption (1), the matrix $L_{\vartheta(t)}$ is symmetric. Hence, there exist an orthonormal $W_{\vartheta(t)}$ such that $W_{\vartheta(t)}^T L_{\vartheta(t)} W_{\vartheta(t)} = \Gamma_{\vartheta(t)} = \text{diag}\{\lambda_{\vartheta(t),1}, \lambda_{\vartheta(t),2}, \ldots, \lambda_{\vartheta(t),N}\}$, in which $\lambda_{\vartheta(t),i}$ are non-negative real numbers.

Moreover, from $\varepsilon(t) = \left[\varepsilon_1^T(t)\varepsilon_2^T(t)\ldots\varepsilon_N^T(t)\right] = (W_{\vartheta(t)}^T \otimes I_n)v(t)$ and $\xi(t) = \left[\xi_1^T(t)\xi_2^T(t)\ldots\xi_N^T(t)\right] = (W_{\vartheta(t)}^T \otimes I_n)\zeta(t)$, the consensus and estimation error systems (15) can be written as

$$
\begin{aligned}
\dot{\varepsilon}(t) &= (I_N \otimes A)\varepsilon(t) + (I_N \otimes A_\tau)\varepsilon(t-\tau) + (W_{\vartheta(t)}^T \otimes I_n)(I_N \otimes E)\Phi(x(t)) \\
&\quad + (\Gamma_{\vartheta(t)} \otimes B(\varphi + \Delta\varphi(t)))\varepsilon(t) + (W_{\vartheta(t)}^T \Xi_{\vartheta(t)} W_{\vartheta(t)} \otimes B(\varphi + \Delta\varphi(t)))\varepsilon(t) \\
&\quad + (W_{\vartheta(t)}^T \Xi_{\vartheta(t)} W_{\vartheta(t)} \otimes B(\varphi + \Delta\varphi(t)))\xi(t) \\
\dot{\xi}(t) &= (I_N \otimes A)\xi(t) + (I_N \otimes A_\tau)\xi(t-\tau) + (W_{\vartheta(t)}^T \otimes I_n)(I_N \otimes E)\Psi(x(t)) \\
&\quad + (I_N \otimes \Theta C)\varepsilon(t) + (I_N \otimes \Theta C)\xi(t) + (\Gamma_{\vartheta(t)} \otimes BK)\xi(t)
\end{aligned}
\tag{16}
$$

Consider the following LKF

$$V(t) = \boldsymbol{\varepsilon}^T(t)(I_N \otimes P)\boldsymbol{\varepsilon}(t) + \xi^T(t)(I_N \otimes Q)\xi(t)$$
$$+ \int_{t-\tau}^t \boldsymbol{\varepsilon}^T(\theta)(I_N \otimes R)\boldsymbol{\varepsilon}(\theta)\mathrm{d}\theta + \int_{t-\tau}^t \xi^T(\theta)(I_N \otimes S)\xi(\theta)\mathrm{d}\theta \tag{17}$$

The time derivative of $V(t)$ yields that

$$
\begin{aligned}
\dot{V}(t) &= \dot{\boldsymbol{\varepsilon}}^T(t)(I_N \otimes P)\boldsymbol{\varepsilon}(t) + \boldsymbol{\varepsilon}^T(t)(I_N \otimes P)\dot{\boldsymbol{\varepsilon}}(t) + \dot{\xi}^T(t)(I_N \otimes Q)\xi(t) \\
&\quad + \xi^T(t)(I_N \otimes Q)\dot{\xi}(t) + \boldsymbol{\varepsilon}^T(t)(I_N \otimes R)\boldsymbol{\varepsilon}(t) - \boldsymbol{\varepsilon}^T(t-\tau)(I_N \otimes R)\boldsymbol{\varepsilon}(t-\tau) \\
&\quad + \xi^T(t)(I_N \otimes S)\xi(t) - \xi^T(t-\tau)(I_N \otimes S)\xi(t-\tau) \\
&= \boldsymbol{\varepsilon}^T(t)(I_N \otimes (A^TP + PA))\boldsymbol{\varepsilon}(t) \\
&\quad + \boldsymbol{\varepsilon}^T(t)(\Gamma_{\vartheta(t)} \otimes (PB(\varphi + \Delta\varphi(t)) + (\varphi + \Delta\varphi(t))^TB^TP))\boldsymbol{\varepsilon}(t) \\
&\quad + \boldsymbol{\varepsilon}^T(t)(W_{\vartheta(t)}^T\Xi_{\vartheta(t)}W_{\vartheta(t)} \otimes (PB(\varphi + \Delta\varphi(t)) + (\varphi + \Delta\varphi(t))^TB^TP))\boldsymbol{\varepsilon}(t) \\
&\quad + 2\boldsymbol{\varepsilon}^T(t)(W_{\vartheta(t)}^T\Xi_{\vartheta(t)}W_{\vartheta(t)} \otimes (PB(\varphi + \Delta\varphi(t))))\xi(t) \\
&\quad + 2\boldsymbol{\varepsilon}^T(t)(W_{\vartheta(t)}^T \otimes PE)\Phi(x(t)) + \xi^T(t)(I_N \otimes (A^TQ + QA))\xi(t) \\
&\quad + \xi^T(t)(\Gamma_{\vartheta(t)} \otimes (QBK + K^TB^TQ))\xi(t) \\
&\quad + \xi^T(t)(I_N \otimes (Q\Theta C + C^T\Theta^TQ))\xi(t) + 2\boldsymbol{\varepsilon}^T(t)(I_N \otimes C^T\Theta^TQ)\xi(t) \\
&\quad + 2\xi^T(t)(W_{\vartheta(t)}^T \otimes QE)\Psi(x(t)) + 2\boldsymbol{\varepsilon}^T(t)(I_N \otimes PA_\tau)\boldsymbol{\varepsilon}(t-\tau) \\
&\quad + 2\xi^T(t)(I_N \otimes QA_\tau)\xi(t-\tau) \\
&\quad + \boldsymbol{\varepsilon}^T(t)(I_N \otimes R)\boldsymbol{\varepsilon}(t) - \boldsymbol{\varepsilon}^T(t-\tau)(I_N \otimes R)\boldsymbol{\varepsilon}(t-\tau) \\
&\quad + \xi^T(t)(I_N \otimes S)\xi(t) - \xi^T(t-\tau)(I_N \otimes S)\xi(t-\tau)
\end{aligned}
\tag{18}
$$

Then, we further decompose (19) as

$$
\begin{aligned}
\dot{V}(t) &= \boldsymbol{\varepsilon}^T(t)(I_N \otimes (A^TP + PA))\boldsymbol{\varepsilon}(t) + \boldsymbol{\varepsilon}^T(t)(\Gamma_{\vartheta(t)} \otimes (PB\varphi + \varphi^TB^TP))\boldsymbol{\varepsilon}(t) \\
&\quad + \boldsymbol{\varepsilon}^T(t)(W_{\vartheta(t)}^T\Xi_{\vartheta(t)}W_{\vartheta(t)} \otimes (PB\varphi + \varphi^TB^TP))\boldsymbol{\varepsilon}(t) \\
&\quad + 2\boldsymbol{\varepsilon}^T(t)(W_{\vartheta(t)}^T\Xi_{\vartheta(t)}W_{\vartheta(t)} \otimes (PB\varphi))\xi(t) \\
&\quad + 2\boldsymbol{\varepsilon}^T(t)(W_{\vartheta(t)}^T \otimes PE)\Phi(x(t)) + \xi^T(t)(I_N \otimes (A^TQ + QA))\xi(t) \\
&\quad + \xi^T(t)(\Gamma_{\vartheta(t)} \otimes (QBK + K^TB^TQ))\xi(t) \\
&\quad + \xi^T(t)(I_N \otimes (Q\Theta C + C^T\Theta^TQ))\xi(t) + 2\boldsymbol{\varepsilon}^T(t)(I_N \otimes C^T\Theta^TQ)\xi(t) \\
&\quad + 2\xi^T(t)(W_{\vartheta(t)}^T \otimes QE)\Psi(x(t)) + 2\boldsymbol{\varepsilon}^T(t)(I_N \otimes PA_\tau)\boldsymbol{\varepsilon}(t-\tau) \\
&\quad + 2\xi^T(t)(I_N \otimes QA_\tau)\xi(t-\tau) \\
&\quad + \boldsymbol{\varepsilon}^T(t)(\Gamma_{\vartheta(t)} \otimes (PB\Delta\varphi(t) + \Delta\varphi^T(t)B^TP))\boldsymbol{\varepsilon}(t) \\
&\quad + \boldsymbol{\varepsilon}^T(t)(W_{\vartheta(t)}^T\Xi_{\vartheta(t)}W_{\vartheta(t)} \otimes (PB\Delta\varphi(t) + \Delta\varphi^T(t)B^TP))\boldsymbol{\varepsilon}(t) \\
&\quad + 2\boldsymbol{\varepsilon}^T(t)(W_{\vartheta(t)}^T\Xi_{\vartheta(t)}W_{\vartheta(t)} \otimes (PB\Delta\varphi(t)))\xi(t) \\
&\quad + \boldsymbol{\varepsilon}^T(t)(I_N \otimes R)\boldsymbol{\varepsilon}(t) - \boldsymbol{\varepsilon}^T(t-\tau)(I_N \otimes R)\boldsymbol{\varepsilon}(t-\tau) \\
&\quad + \xi^T(t)(I_N \otimes S)\xi(t) - \xi^T(t-\tau)(I_N \otimes S)\xi(t-\tau)
\end{aligned}
\tag{19}
$$

Based on the Lemma 1 and Lipschitz condition (2), and properties of the Kronecker product, it can be deduced that

$$
\begin{aligned}
\dot{V}(t) \quad &\leq \varepsilon^T(t)(I_N \otimes (A^T P + PA))\varepsilon(t) + \varepsilon^T(t)(\Gamma_{\vartheta(t)} \otimes (PB\varphi + \varphi^T B^T P))\varepsilon(t) \\
&+ (1 + \epsilon_3)\varepsilon^T(t)((W_{\vartheta(t)}^T \otimes \varphi^T)(\Xi_{\vartheta(t)} \otimes I_m)(W_{\vartheta(t)} \otimes B^T P) \\
&+ (W_{\vartheta(t)}^T \otimes PB)(\Xi_{\vartheta(t)} \otimes I_m)(W_{\vartheta(t)} \otimes \varphi))\varepsilon(t) \\
&+ \epsilon_1 \varepsilon^T(t)(I_N \otimes PEE^T P)\varepsilon(t) + \epsilon_1^{-1}\gamma^2 \varepsilon^T(t)\varepsilon(t) \\
&+ \xi^T(t)(I_N \otimes (A^T Q + QA))\xi(t) \\
&+ \xi^T(t)(\Gamma_{\vartheta(t)} \otimes (QBK + K^T B^T Q))\xi(t) \\
&+ \xi^T(t)(I_N \otimes (Q\Theta C + C^T \Theta^T Q))\xi(t) + 2\varepsilon^T(t)(I_N \otimes C^T \Theta^T Q)\xi(t) \\
&+ \epsilon_2 \xi^T(t)(I_N \otimes QEE^T Q)\xi(t) + (\epsilon_2^{-1}\gamma^2 + \epsilon_3^{-1} + \epsilon_4^{-1})\xi^T(t)\xi(t) \\
&+ 2\varepsilon^T(t)(I_N \otimes PA_\tau)\varepsilon(t-\tau) \\
&+ 2\xi^T(t)(I_N \otimes QA_\tau)\xi(t-\tau) \\
&+ \varepsilon^T(t)(\Gamma_{\vartheta(t)} \otimes (PB\Delta\varphi(t) + \Delta\varphi^T(t)B^T P))\varepsilon(t) \\
&+ \varepsilon^T(t)((I_N \otimes PBM)(\Gamma_{\vartheta(t)} \otimes \Delta\varphi(t))(I_N \otimes N) + \\
&(I_N \otimes N^T)(\Gamma_{\vartheta(t)} \otimes \Delta\varphi^T(t))(I_N \otimes M^T B^T P))\varepsilon(t) \\
&+ (1 + \epsilon_4)\varepsilon^T(t)((W_{\vartheta(t)}^T \otimes PBM)(\Xi_{\vartheta(t)} \otimes \Delta\varphi(t))(W_{\vartheta(t)} \otimes N) + \\
&(W_{\vartheta(t)}^T \otimes N^T)(\Xi_{\vartheta(t)} \otimes \Delta\varphi^T(t))(W_{\vartheta(t)} \otimes M^T B^T P))\varepsilon(t) \\
&+ \varepsilon^T(t)(I_N \otimes R)\varepsilon(t) - \varepsilon^T(t-\tau)(I_N \otimes R)\varepsilon(t-\tau) \\
&+ \xi^T(t)(I_N \otimes S)\xi(t) - \xi^T(t-\tau)(I_N \otimes S)\xi(t-\tau)
\end{aligned}
\tag{20}
$$

In (20), we have an unknown parameter due to the existence of the parametric uncertain matrix (norm-bounded uncertainty in the gain of controller). To tackle this issue and achieve an upper bound, and convert the problem and compute the gains in terms of LMIs for the stability analysis, we use Lemma 2. Moreover, by defining $\varphi = B^T P$ and $K = B^T Q$, one can acquire:

$$
\begin{aligned}
\dot{V}(t) \quad &\leq \varepsilon^T(t)(I_N \otimes (A^T P + PA))\varepsilon(t) + 2\varepsilon^T(t)(\Gamma_{\vartheta(t)} \otimes (PBB^T P)\varepsilon(t) \\
&+ (1 + \epsilon_3)(\alpha_3^{-1} + \alpha_3 \mu^2)\varepsilon^T(t)(I_N \otimes PBB^T P)\varepsilon(t) \\
&+ \epsilon_1 \varepsilon^T(t)(I_N \otimes PEE^T P)\varepsilon(t) + \epsilon_1^{-1}\gamma^2 \varepsilon^T(t)\varepsilon(t) \\
&+ \xi^T(t)(I_N \otimes (A^T Q + QA))\xi(t) \\
&+ 2\xi^T(t)(\Gamma_{\vartheta(t)} \otimes (QBB^T Q))\xi(t) \\
&+ \xi^T(t)(I_N \otimes (Q\Theta C + C^T \Theta^T Q))\xi(t) + 2\varepsilon^T(t)(I_N \otimes C^T \Theta^T Q)\xi(t) \\
&+ \epsilon_2 \xi^T(t)(I_N \otimes QEE^T Q)\xi(t) + (\epsilon_2^{-1}\gamma^2 + \epsilon_3^{-1} + \epsilon_4^{-1})\xi^T(t)\xi(t) \\
&+ 2\varepsilon^T(t)(I_N \otimes PA_\tau)\varepsilon(t-\tau) \\
&+ 2\xi^T(t)(I_N \otimes QA_\tau)\xi(t-\tau) \\
&+ (\alpha_2^{-1} + (1 + \epsilon_4)\alpha_1^{-1})\varepsilon^T(t)(I_N \otimes PBMM^T B^T P)\varepsilon(t) \\
&+ (\alpha_2 \lambda^2 + (1 + \epsilon_4)\alpha_1 \mu^2)\varepsilon^T(t)(I_N \otimes N^T N)\varepsilon(t) \\
&+ \varepsilon^T(t)(I_N \otimes R)\varepsilon(t) - \varepsilon^T(t-\tau)(I_N \otimes R)\varepsilon(t-\tau) \\
&+ \xi^T(t)(I_N \otimes S)\xi(t) - \xi^T(t-\tau)(I_N \otimes S)\xi(t-\tau)
\end{aligned}
\tag{21}
$$

Therefore, considering (21), based on the definitions of the Kronecker product, and definition of $\lambda$ and $\mu$ which are proposed in the Theorem 1, one can acquire the upper bound based on the dynamics of consensus and estimation errors. Then, one can infer that

$$
\begin{aligned}
\dot{V}(t) \leq & \sum_{i=1}^{N} \varepsilon_i^T(t)(A^T P + PA + (2\lambda + (1+\epsilon_3)(\alpha_3^{-1} + \alpha_3\mu^2))PBB^T P \\
& + \epsilon_1 PEE^T P + \epsilon_1^{-1}\gamma^2 I + R + (\alpha_2^{-1} + (1+\epsilon_4)\alpha_1^{-1})PBMM^T B^T P \\
& (\alpha_2\lambda^2 + (1+\epsilon_4)\alpha_1\mu^2)N^T N)\varepsilon_i(t) \\
& + \xi_i^T(t)(A^T Q + QA + 2\lambda QBB^T Q + Q\Theta C + C^T\Theta^T Q + \epsilon_2 QEE^T Q \\
& + (\epsilon_2^{-1}\gamma^2 + \epsilon_3^{-1} + \epsilon_4^{-1})I + S)\xi_i(t) + 2\varepsilon_i^T(t)(C^T\Theta^T Q)\xi_i(t) \\
& + 2\varepsilon_i^T(t)PA_\tau\varepsilon_i(t-\tau) + 2\xi_i^T(t)QA_\tau\xi_i(t-\tau) \\
& + \varepsilon_i^T(t-\tau)(-R)\varepsilon_i(t-\tau) + \xi_i^T(t-\tau)(-S)\xi_i(t-\tau) \\
= & \sum_{i=1}^{N} \varphi_i^T(t)\Pi\varphi_i(t)
\end{aligned}
\tag{22}
$$

where $\phi_i(t) = \left[\varepsilon_i^T(t)\xi_i^T(t)\varepsilon_i^T(t-\tau)\xi_i^T(t-\tau)\right]^T$ and

$$
\begin{aligned}
\Pi = & \begin{bmatrix}
\Pi_{1,1} & C^T\Theta^T Q^T & PA_\tau & 0 \\
* & \Pi_{2,2} & 0 & QA_\tau \\
* & * & -R & 0 \\
* & * & * & -S
\end{bmatrix} \\[10pt]
\Pi_{1,1} = & A^T P + PA + (2\lambda + (1+\epsilon_3)(\alpha_3^{-1} + \alpha_3\mu^2))PBB^T P \\
& + \epsilon_1 PEE^T P + \epsilon_1^{-1}\gamma^2 I + R + (\alpha_2^{-1} + (1+\epsilon_4)\alpha_1^{-1})PBMM^T B^T P \\
& + (\alpha_2\lambda^2 + (1+\epsilon_4)\alpha_1\mu^2)N^T N \\
\Pi_{2,2} = & A^T Q + QA + 2\lambda QBB^T Q + Q\Theta C + C^T\Theta^T Q + \epsilon_2 QEE^T Q \\
& + (\epsilon_2^{-1}\gamma^2 + \epsilon_3^{-1} + \epsilon_4^{-1})I + S
\end{aligned}
\tag{23}
$$

Applying Schur Complement Lemma and using the change of variable $Q\Theta = X$, the constraint $\Pi < 0$ is converted to the LMI (10).

Therefore, one can conclude that the followers in (1) asymptotically observe the leader (3) via (7) and the NMAS achieves the LF consensus by employing the protocol (8). Therefore, the proof is completed.

**Remark 2**. *Although a distributed controller and observer under any arbitrary switching signals, determining the active communication graph topology, is designed in this paper, the proposed protocols can apply to the system under a switching signal satisfying the specific dwell-time.*

**Remark 3**. *Since the robustness of the proposed controller is investigated, the consensus protocol (8) can be used in many practical operations when the actuator of every agent i suffers from the attack. Therefore, the strategy of this paper can study resilient consensus of NMASs under actuator attacks.*

## $H_\infty$ Leader-following consensus

Sufficient conditions in terms of LMI are achieved for extending the LF consensus protocol (8) to ensure the $H_\infty$ disturbance attenuation level.

**Theorem 2**. *Consider the network of agents* (1) *and* (3) *with time-delay, utilizing DSO scheme of* (7) *and the control policy* (8) *under switching graphs* $\sigma_{9(t)}$. *For given positive scalars* $\epsilon_1$ : $\epsilon_4$, $\alpha_1$ : $\alpha_3$, $\beta$, *suppose that symmetric matrices* $P > 0$, $Q > 0$, $R > 0$, $S > 0$, *and matrix* $X$ *with suitable dimensions can be found such that*

$$
\begin{bmatrix}
\Omega_{1,1} & C^T X^T & PA_\tau & 0 & \sqrt{\psi_1}PB & \sqrt{\epsilon_1}PE & \sqrt{\psi_2}PBM & 0 & 0 & PW \\
* & \Omega_{2,2} & 0 & QA_\tau & 0 & 0 & 0 & \sqrt{2\lambda}QB & \sqrt{2\lambda}QE & 0 \\
* & * & -R & 0 & 0 & 0 & 0 & 0 & 0 & 0 \\
* & * & * & -S & 0 & 0 & 0 & 0 & 0 & 0 \\
* & * & * & * & -I & 0 & 0 & 0 & 0 & 0 \\
* & * & * & * & * & -I & 0 & 0 & 0 & 0 \\
* & * & * & * & * & * & -I & 0 & 0 & 0 \\
* & * & * & * & * & * & * & -I & 0 & 0 \\
* & * & * & * & * & * & * & * & -I & 0 \\
* & * & * & * & * & * & * & * & * & -\beta^2 I
\end{bmatrix}
$$

$$< 0$$

$$
\begin{aligned}
\Omega_{1,1} &= A^T P + PA + (\epsilon_1^{-1}\gamma^2 + 1)I + R + (\alpha_2\lambda^2 + (1 + \epsilon_4)\alpha_1\mu^2)N^T N \\
\Omega_{2,2} &= A^T Q + QA + XC + C^T X^T + (\epsilon_2^{-1}\gamma^2 + \epsilon_3^{-1} + \epsilon_4^{-1})I + S \\
\psi_1 &= 2\lambda + (1 + \epsilon_3)(\alpha_3^{-1} + \alpha_3\mu^2), \quad \psi_2 = \alpha_2^{-1} + (1 + \epsilon_4)\alpha_1^{-1}
\end{aligned}
\tag{24}
$$

*Then, the followers observe the leader* (3) *and the NMAS achieves the guaranteed-performance* $H_\infty$ *LF consensus. Moreover the feedback gain of the control policy* (8) *is achieved as* $\varphi = B^T P$ *and the DSO gains in* (7) *are computed via* $K = B^T Q$ *and* $\Theta = Q^{-1} X$.

*Proof.* Denoting $d(t) = \left[ d_1^T(t) d_2^T(t) \dots d_N^T(t) \right]^T$ and considering $d(t) \neq 0$, the term $(I_N \otimes W)$ $d(t)$ will be added to $\dot{\upsilon}(t)$ dynamics in (15). Now, using Lemma 1, inequality (21) is satisfied if

the following inequality holds

$$
\begin{aligned}
\dot{V}(t) \quad &\leq \boldsymbol{\varepsilon}^T(t)(I_N \otimes (A^T P + PA))\boldsymbol{\varepsilon}(t) + 2\boldsymbol{\varepsilon}^T(t)(\Gamma_{\vartheta(t)} \otimes (PBB^T P)\boldsymbol{\varepsilon}(t) \\
&\quad + (1 + \epsilon_3)(\alpha_3^{-1} + \alpha_3 \mu^2)\boldsymbol{\varepsilon}^T(t)(I_N \otimes PBB^T P)\boldsymbol{\varepsilon}(t) \\
&\quad + \epsilon_1 \boldsymbol{\varepsilon}^T(t)(I_N \otimes PEE^T P)\boldsymbol{\varepsilon}(t) + \epsilon_1^{-1}\gamma^2\boldsymbol{\varepsilon}^T(t)\boldsymbol{\varepsilon}(t) \\
&\quad + \xi^T(t)(I_N \otimes (A^T Q + QA))\xi(t) \\
&\quad + 2\xi^T(t)(\Gamma_{\vartheta(t)} \otimes (QBB^T Q))\xi(t) \\
&\quad + \xi^T(t)(I_N \otimes (Q\Theta C + C^T \Theta^T Q))\xi(t) + 2\boldsymbol{\varepsilon}^T(t)(I_N \otimes C^T \Theta^T Q)\xi(t) \\
&\quad + \epsilon_2 \xi^T(t)(I_N \otimes QEE^T Q)\xi(t) + (\epsilon_2^{-1}\gamma^2 + \epsilon_3^{-1} + \epsilon_4^{-1})\xi^T(t)\xi(t) \\
&\quad + 2\boldsymbol{\varepsilon}^T(t)(I_N \otimes PA_\tau)\boldsymbol{\varepsilon}(t - \tau) \\
&\quad + 2\xi^T(t)(I_N \otimes QA_\tau)\xi(t - \tau) \\
&\quad + (\alpha_2^{-1} + (1 + \epsilon_4)\alpha_1^{-1})\boldsymbol{\varepsilon}^T(t)(I_N \otimes PBMM^T B^T P)\boldsymbol{\varepsilon}(t) \\
&\quad + (\alpha_2\lambda^2 + (1 + \epsilon_4)\alpha_1\mu^2)\boldsymbol{\varepsilon}^T(t)(I_N \otimes N^T N)\boldsymbol{\varepsilon}(t) \\
&\quad + \boldsymbol{\varepsilon}^T(t)(I_N \otimes R)\boldsymbol{\varepsilon}(t) - \boldsymbol{\varepsilon}^T(t - \tau)(I_N \otimes R)\boldsymbol{\varepsilon}(t - \tau) \\
&\quad + \xi^T(t)(I_N \otimes S)\xi(t) - \xi^T(t - \tau)(I_N \otimes S)\xi(t - \tau) \\
&\quad + \boldsymbol{\varepsilon}^T(t)(I_N \otimes \beta^{-2} PWW^T P)\boldsymbol{\varepsilon}(t) + \beta^2 d^T(t)d(t)
\end{aligned}
\tag{25}
$$

Concerning the $H_\infty$ performance of the analogous consensus error system (15), $\mathcal{J} = \int_0^\infty (v^T(t)v(t) - \beta^2 d^T(t)d(t))\mathrm{dt}$ is designated as a cost function, and using nonsingular transformation one can get that

$$
\mathcal{J} = \int_0^\infty (\boldsymbol{\varepsilon}^T(t)\boldsymbol{\varepsilon}(t) - \beta^2 d^T(t)d(t))\mathrm{dt}
\tag{26}
$$

Regarding (25) and considering zero initial condition, it can be acquired that

$$
\begin{aligned}
\mathcal{J} &= \int_0^\infty (\boldsymbol{\varepsilon}^T(t)\boldsymbol{\varepsilon}(t) - \beta^2 d^T(t)d(t) + \dot{V}(t))\mathrm{dt} - V(\infty) + V(0) \\
&\leq \int_0^\infty (\boldsymbol{\varepsilon}^T(t)\boldsymbol{\varepsilon}(t) - \beta^2 d^T(t)d(t) + \dot{V}(t))\mathrm{dt} \\
&\leq \sum_{i=1}^N \int_0^\infty (\boldsymbol{\varepsilon}_i^T(t)(A^T P + PA + (2\lambda + (1 + \epsilon_3)(\alpha_3^{-1} + \alpha_3\mu^2))PBB^T P \\
&\quad + \epsilon_1 PEE^T P + (\epsilon_1^{-1}\gamma^2 + 1)I + R + (\alpha_2^{-1} + (1 + \epsilon_4)\alpha_1^{-1})PBMM^T B^T P \\
&\quad (\alpha_2\lambda^2 + (1 + \epsilon_4)\alpha_1\mu^2)N^T N + \beta^{-2}PWW^T P)\boldsymbol{\varepsilon}_i(t) \\
&\quad + \xi_i^T(t)(A^T Q + QA + 2\lambda QBB^T Q + Q\Theta C + C^T \Theta^T Q + \epsilon_2 QEE^T Q \\
&\quad + (\epsilon_2^{-1}\gamma^2 + \epsilon_3^{-1} + \epsilon_4^{-1})I + S)\xi_i(t) + 2\boldsymbol{\varepsilon}_i^T(t)(C^T \Theta^T Q)\xi_i(t) \\
&\quad + 2\boldsymbol{\varepsilon}_i^T(t)PA_\tau \boldsymbol{\varepsilon}_i(t - \tau) + 2\xi_i^T(t)QA_\tau \xi_i(t - \tau) \\
&\quad + \boldsymbol{\varepsilon}_i^T(t - \tau)(-R)\boldsymbol{\varepsilon}_i(t - \tau) + \xi_i^T(t - \tau)(-S)\xi_i(t - \tau))\mathrm{dt} \\
&= \sum_{i=1}^N \int_0^\infty \varphi_i^T(t)\Upsilon\varphi_i(t)\mathrm{dt}
\end{aligned}
\tag{27}
$$

where

$$\Upsilon = \begin{bmatrix} \Upsilon_{1,1} & C^T\Theta^TQ^T & PA_\tau & 0 \\ * & \Upsilon_{2,2} & 0 & QA_\tau \\ * & * & -R & 0 \\ * & * & * & -S \end{bmatrix} \qquad (28)$$

$$\begin{aligned}
\Upsilon_{1,1} &= A^TP + PA + (2\lambda + (1+\epsilon_3)(\alpha_3^{-1} + \alpha_3\mu^2))PBB^TP \\
&\quad + \epsilon_1 PEE^TP + (\epsilon_1^{-1}\gamma^2 + 1)I + R + (\alpha_2^{-1} + (1+\epsilon_4)\alpha_1^{-1})PBMM^TB^TP \\
&\quad + (\alpha_2\lambda^2 + (1+\epsilon_4)\alpha_1\mu^2)N^TN + \beta^{-2}PWW^TP \\
\Upsilon_{2,2} &= A^TQ + QA + 2\lambda QBB^TQ + Q\Theta C + C^T\Theta^TQ + \epsilon_2 QEE^TQ \\
&\quad + (\epsilon_2^{-1}\gamma^2 + \epsilon_3^{-1} + \epsilon_4^{-1})I + S
\end{aligned}$$

Applying Schur Complement Lemma and based on the change of variable $Q\Theta = X$, the constraint $Y < 0$ can be converted to the equivalent form of LMI (24) which implies that

$$\mathcal{J} = \int_0^\infty (v^T(t)v(t) - \beta^2 d^T(t)d(t))\mathrm{dt} < 0 \qquad (29)$$

From (29), it is obvious that $\|v(t)\| < \beta\|d(t)\|$ holds for any $d(t) \neq 0$, in which $d(t) \in l_2[0, +\infty); \mathbb{R}^{qN}$. Therefore, the followers in (1) observe the leader (3) via (7) and the NMAS achieves the ensured-performance $H_\infty$ LF consensus by employing (8). This completes the proof.

**Remark 4**. The parameter $\beta$ is related to the cost function (26) and we try to reduce it in order to reduce the effect of external disturbances on the consensus error dynamics. Moreover, there is no restriction in the choice of $\alpha$, since they can enable us to tackle the feasibility of the problem/LMIs.

## Simulations

In this section, a numerical study is simulated to assess the suggested strategy.

Consider the network of NMAS incorporating one leader and 5 followers. The interaction graph is depicted in Fig 3 and the switching signal $\vartheta(t)$ that determines the active graph topology is demonstrated in Fig 4.

Although the spanning tree is not included in each graph $\sigma_\iota(\iota = 1, 2, \ldots, \ell)$, it exists in the union $\bar{\sigma} = \bigcup_{\iota=1}^\ell \sigma_\iota$. In addition, the weight edges is considered to be 1. The system parameters

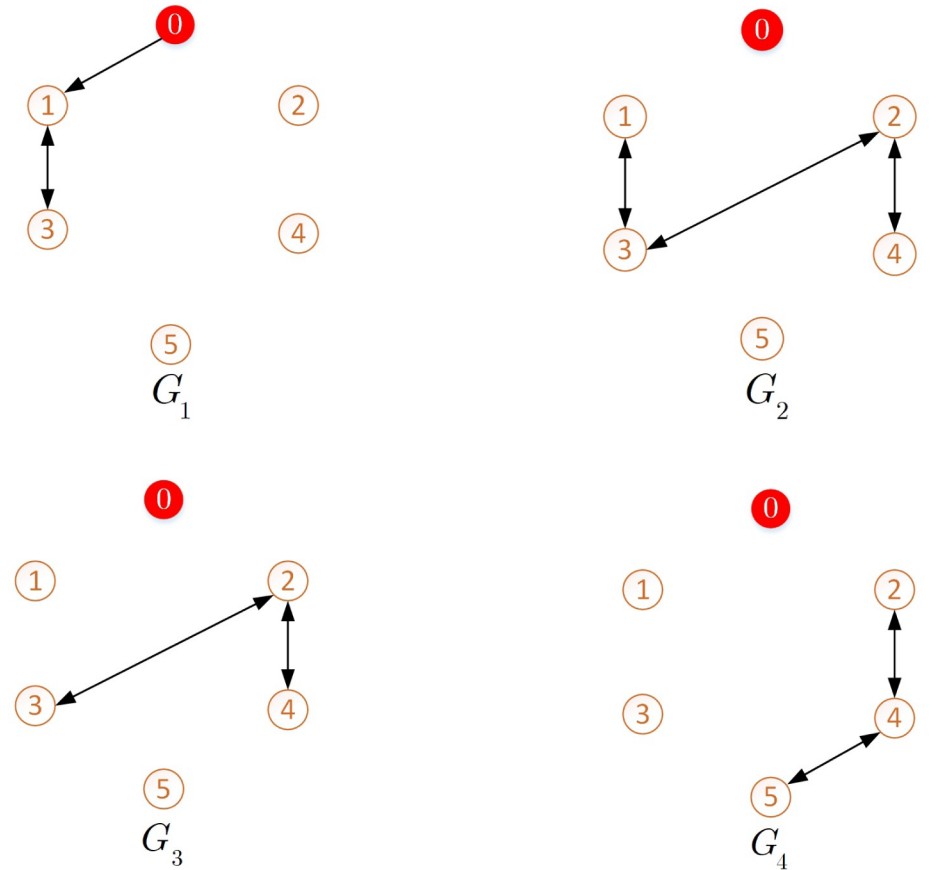

**Fig 3. Switching topologies of the graphs.**

are given as

$$
\begin{aligned}
A &= \begin{bmatrix} 0 & 1 & 0 \\ -3 & -2 & 1 \\ 0 & 0 & -1 \end{bmatrix}, A_\tau = \begin{bmatrix} 0.2 & 0.1 & 0 \\ -0.1 & 0.2 & 0.2 \\ 0.2 & 0.1 & -0.2 \end{bmatrix}, A_n = \begin{bmatrix} 0.06 & 0.01 & 0.02 \\ 0.01 & 0.02 & 0.06 \\ 0.05 & 0.02 & 0.02 \end{bmatrix} \\
E &= \begin{bmatrix} 1 & 0 & 0 \\ 0 & 1 & 0 \\ 0 & 0 & 1 \end{bmatrix}, B = \begin{bmatrix} 0 \\ 0.1 \\ 0 \end{bmatrix}, W = \begin{bmatrix} 1.6 & 0.3 & 0.8 \\ 0.1 & 0.5 & 1 \\ 0.7 & 0.4 & 0.2 \end{bmatrix} \\
f(x_i(t)) &= \begin{bmatrix} 0 \\ 0 \\ -0.333\sin(x_i(t)) \end{bmatrix}, f(x_0(t)) = \begin{bmatrix} 0 \\ 0 \\ -0.333\sin(x_0(t)) \end{bmatrix} \\
C &= \begin{bmatrix} 0.6 & 0.8 & 0.5 \end{bmatrix}, C_n = \begin{bmatrix} 0.01 & 0.02 & 0.01 \end{bmatrix}, d_i(t) = \begin{bmatrix} e^{-2t} \\ \sin(t) \\ e^{-6t}\sin(2t) \end{bmatrix}
\end{aligned}
\tag{30}
$$

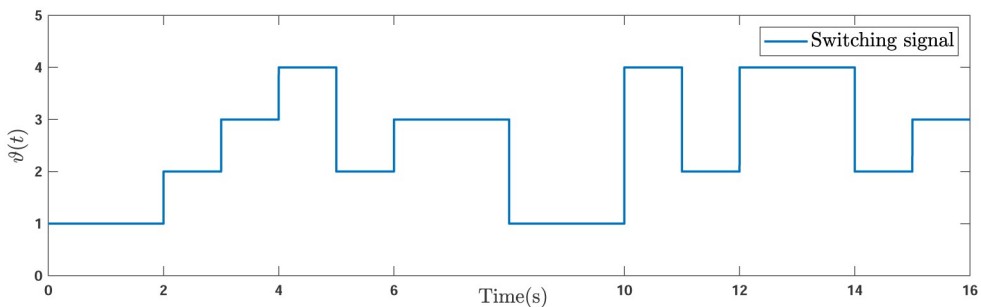

**Fig 4. Switching signal of the simulations.**

In addition, $r(t) : \mathcal{N}(0, 0.06I_{3\times3})$ and $v_i(t) : \mathcal{N}(0, 0.04I_{3\times3})$. The time-delay is selected as $\tau$ = 200(ms), and one can achieve that $\lambda = 3$, $\mu = 1$. It is to be noted that the Lipschitz constant and design parameters are prescribed as $\varrho = 0.333$, $\alpha_1 = 0.6$, $\alpha_2 = 0.7$, $\alpha_3 = 0.5$, $\epsilon_1 = 6$, $\epsilon_2 = 0.1$, and $\epsilon_3 = \epsilon_4 = 2$.

The first aim is to validate distributed observer (7) and controller (8) in the case of $d_i(t) = 0$. By solving the LMI of Theorem 1, one gets the feasible solution of LMI (10) as

$$
\begin{aligned}
P &= \begin{bmatrix} 0.2610 & -0.0236 & -0.0012 \\ -0.0236 & 0.5893 & -0.0745 \\ -0.0012 & -0.0745 & 0.5326 \end{bmatrix}, \quad
Q = \begin{bmatrix} 2.0229 & -1.7578 & 0.0377 \\ -1.7578 & 5.7965 & 0.8021 \\ 0.0377 & 0.8021 & 3.6408 \end{bmatrix} \\
R &= \begin{bmatrix} 4.3030 & -2.5023 & 1.2936 \\ -2.5023 & 6.5657 & 0.1353 \\ 1.2936 & 0.1353 & 4.4340 \end{bmatrix}, \quad
S = \begin{bmatrix} 1.3956 & -0.0789 & 0.1137 \\ -0.0789 & 3.5217 & -0.2872 \\ 0.1137 & -0.2872 & 2.9247 \end{bmatrix} \quad (31) \\
X &= \begin{bmatrix} -1.6848 \\ -1.3161 \\ -1.0911 \end{bmatrix}
\end{aligned}
$$

The parameters involved in the controller gain $\Delta\varphi(t)$ in (9), considered as parametric uncertainty, are selected as $M = 0.2$, $N = [0.2 \ 0.1 \ -0.2]$. Regarding Theorem 1, the observer and controller gains applied in (7) and (8) are computed as follows

$$
\begin{aligned}
\varphi = B^T P &= \begin{bmatrix} -0.0024 & 0.0589 & -0.0075 \end{bmatrix} \\
K = B^T Q &= \begin{bmatrix} -0.1758 & 0.5796 & 0.0802 \end{bmatrix} \\
\Theta = Q^{-1} X &= \begin{bmatrix} -1.3708 \\ -0.6222 \\ -0.1484 \end{bmatrix}
\end{aligned} \quad (32)
$$

During the simulations, the time-varying gain perturbations of the consensus (8) is chosen as $\Delta\varphi(t) = 0.5\cos(t)$. The errors of the consensus are depicted in Figs 5–7 while Figs 8–10 demonstrate the errors of the estimations of the leader's states. Figs 5–7 exhibit the convergence of

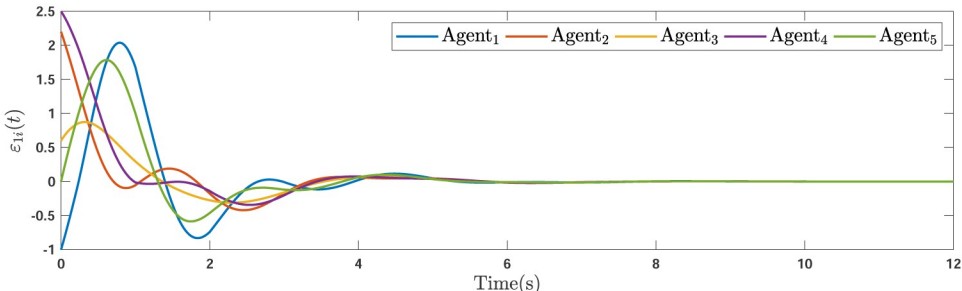

**Fig 5. The first consensus errors of the agents.**

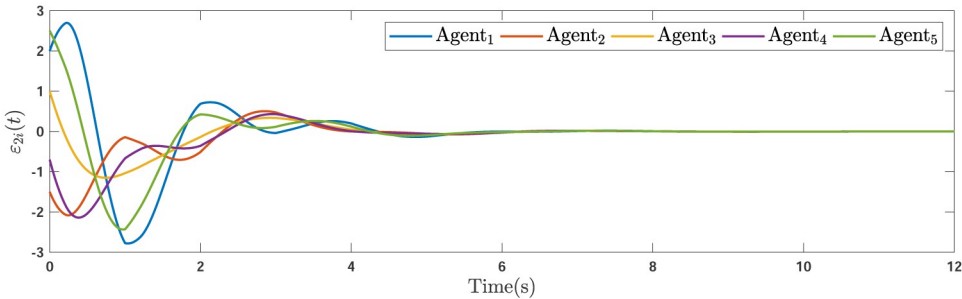

**Fig 6. The second consensus errors of the agents.**

the states of followers to the corresponding leader by applying the consensus protocol (8). It is observed from Figs 8–10 that utilizing the DSO (7), the estimates of the leader's states asymptotically track the leaders states under the influences of time-delay and switching topology.

The value of time-delay in this example is due to the feasibility of LMIs. For the practical example, the value of time-delay is adjusted based on the physical structure of the model. It should be noted that for a given time-delay the free parameters are obtained such that the LMIs to be satisfied.

For the second objective, it is considered to authenticate the results of the Theorem 2 to study the $H_\infty$ LF consensus problem. Let $\beta = 0.5$, then solving LMI (24) leads to the gains of

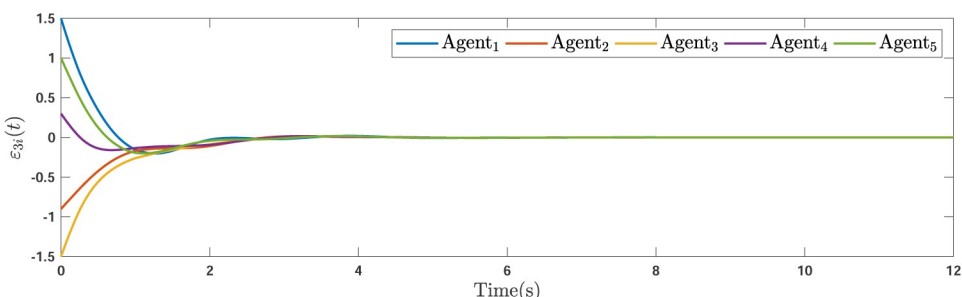

**Fig 7. The third consensus errors of the agents.**

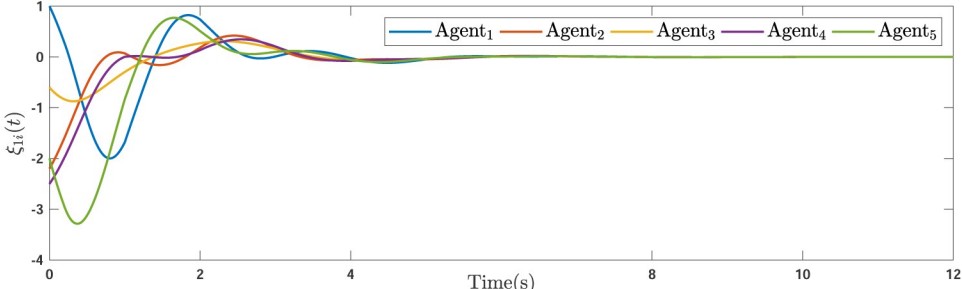

**Fig 8. The first estimation errors.**

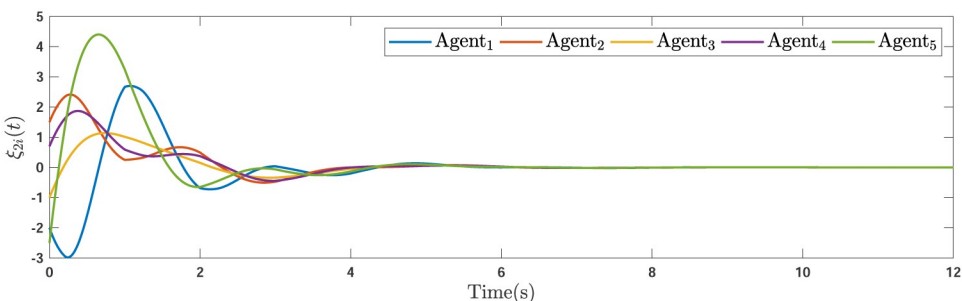

**Fig 9. The second estimation errors.**

DSO (7) and controller (7) as follows

$$
\begin{aligned}
\varphi = B^T P & = \begin{bmatrix} -0.0080 & 0.0635 & -0.0096 \end{bmatrix} \\
K = B^T Q & = \begin{bmatrix} -0.1780 & 0.8793 & 0.1427 \end{bmatrix} \\
\Theta = Q^{-1} X & = \begin{bmatrix} -3.3560 \\ -1.1479 \\ -0.2087 \end{bmatrix}
\end{aligned}
\tag{33}
$$

Applying the distributed consensus protocol (8) and DSO (7) to the system in the presence of exogenous disturbances gives rise to appropriate consensus and estimation errors. Figs 11–

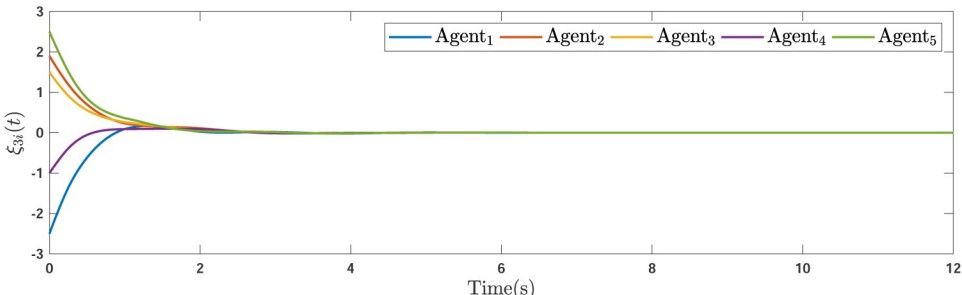

**Fig 10. The third estimation errors.**

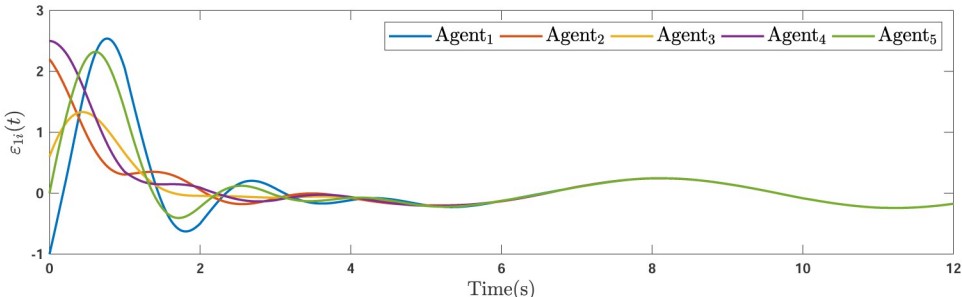

**Fig 11. The first consensus errors of the agents in the presence of external disturbance.**

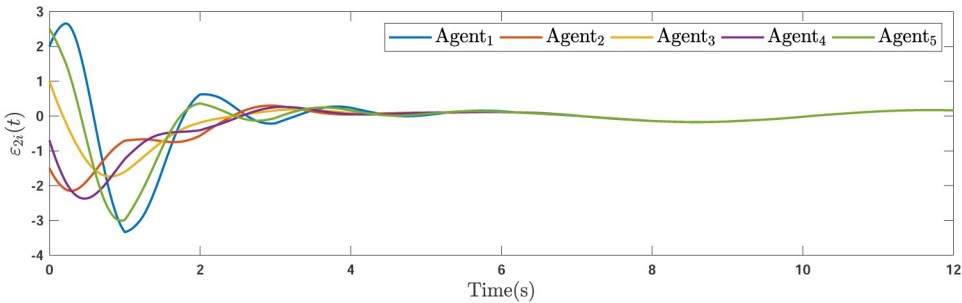

**Fig 12. The second consensus errors of the agents in the presence of external disturbance.**

13 indicates the errors of the consensus while Figs 14–16 illustrate the errors of the estimations of the leader's states in the presence of external disturbance. From Figs 11–13, one can perceive that trajectories of followers well converge by applying the consensus protocol (8). Figs 14–16 reveal that employing the suggested observer (7) leads to a well estimation under the influences of time-delay, external disturbance, and switching graph topology.

To better show the leader's states estimation accuracy and also consensus performance, a comparison with similar approaches are provided. The values of root-mean-square-errors for SMC [41], L2-L$_\infty$ [42] and proposed approach are given in Table 1. The results of Table 1, clearly show the superiority of the designed approach.

**Remark 5**. The results in Figs 3–16, show that the suggested controller well handles the time-varying gain perturbations, and the designed observer well estimates the leader's states.

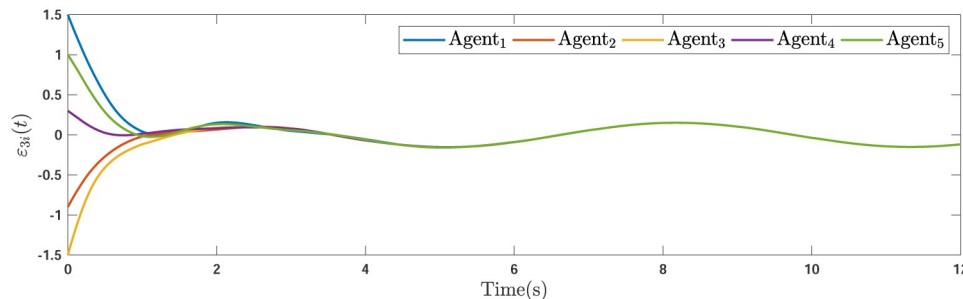

**Fig 13. The third consensus errors of the agents in the presence of external disturbance.**

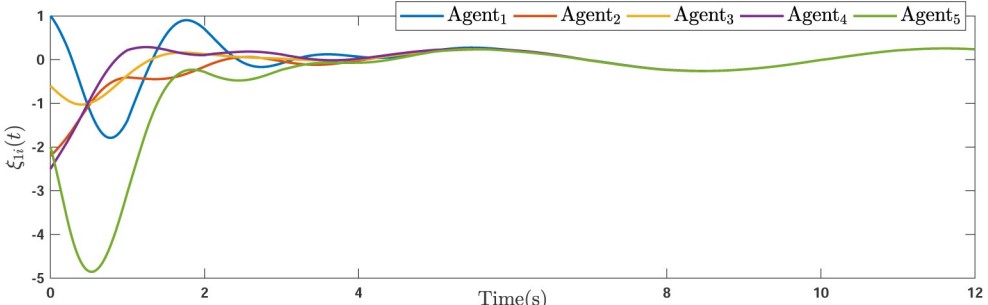

**Fig 14. The first estimation errors of the DSO of agents under external disturbance.**

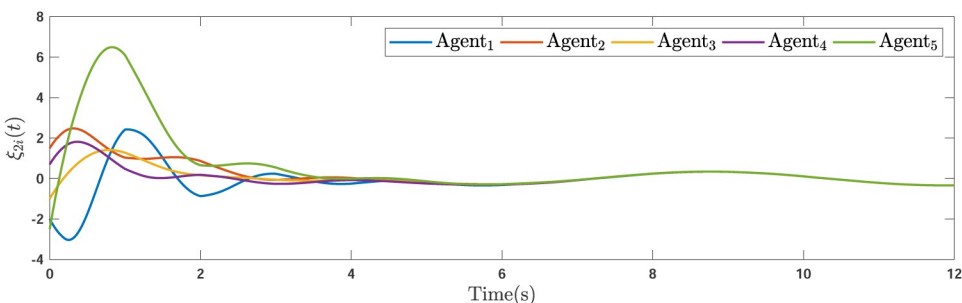

**Fig 15. The second estimation errors of the DSO of agents under external disturbance.**

In addition to perturbations, it is seen that the effect of time-delay and switching topology is well tackled. Furthermore, the effects of external disturbances are handled by the designed $H_\infty$-based scheme. It is demonstrated that the suggested consensus scenario, is good effective in the presence of external disturbance and other perturbations such as time-delay and switching topology. A suitable and desired convergence to zero level is seen in the error trajectories.

**Remark 6**. It should be noted that the distributed observer is designed to estimate the leader's states, and based on the estimation, the distributed controller is applied to the system to solve the consensus problem. Therefore, the role of the observer with stability analysis of the estimation error (error between the states of the leader and its estimations) is analyzed. To this

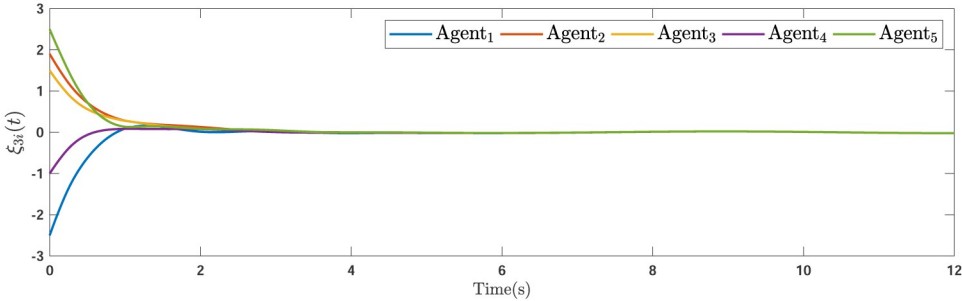

**Fig 16. The third estimation errors of the DSO of agents under external disturbance.**

**Table 1. Comparisons of RMSEs.**

| Error | Proposed | SMC [41] | L2-L$_\infty$ [42] |
|---|---|---|---|
| $\varepsilon_{11}$ | 0.4748 | 0.8701 | 0.4801 |
| $\varepsilon_{12}$ | 0.3075 | 0.5407 | 0.4504 |
| $\varepsilon_{13}$ | 0.2250 | 0.2147 | 0.6327 |
| $\varepsilon_{14}$ | 0.4133 | 0.7414 | 0.5447 |
| $\varepsilon_{15}$ | 0.4261 | 0.5784 | 0.6014 |

end, Lyapunov-Krasovskii functional approach is considered and it is guaranteed that the error converges to the origin.

Remark 7. It is worth recalling that the most important practical restrictions such as time-delay, external disturbances, time-varying gain perturbations and switching topologies, have been considered in the control design. Then, the suggested approach can be easily used in real-world applications. For our future studies, the designed approach will be applied to a group of searching robots.

## Conclusion

In this paper, a DSO is presented to estimate the leader states in a class of nonlinear MASs. Also, the switching graph topologies are investigated. Designing a distributed controller, the LF consensus problem under the influences of time-delay in leader's and followers' states is studied. Thanks to an appropriate LKF along with algebraic graph theory, sufficient conditions in terms of LMI are acquired and solved to ensure the stability of the suggested distributed observer and controller. The robustness of the proposed distributed control protocol against gain perturbations is ensured. Furthermore, considering a prescribed $H_\infty$ disturbance attenuation level, a robust $H_\infty$ distributed control policy is extended to preserve the robust performance of the system against external disturbances. The feasibility of the LMI constraints and efficiency of the proposed distributed algorithm are admitted and demonstrated through simulation results. It should be noted that the choice of design parameters is still a challenging problem. For future studies, modern techniques such as neuro-fuzzy approaches can be developed for better tuning of these design parameters.

## Acknowledgments

The author would like to thank Dr. Rabia Safdar for her contribution on this paper.

## Author Contributions

**Conceptualization:** Amin Taghieh, Ardashir Mohammadzadeh, Sami ud Din, Wudhichai Assawinchaichote, Afef Fekih.

**Data curation:** Wudhichai Assawinchaichote, Afef Fekih.

**Formal analysis:** Amin Taghieh, Sami ud Din, Afef Fekih.

**Investigation:** Sami ud Din, Saleh Mobayen, Wudhichai Assawinchaichote.

**Methodology:** Ardashir Mohammadzadeh, Sami ud Din, Saleh Mobayen, Wudhichai Assawinchaichote.

**Resources:** Wudhichai Assawinchaichote.

**Supervision:** Ardashir Mohammadzadeh, Saleh Mobayen.

**Validation:** Ardashir Mohammadzadeh, Saleh Mobayen.

**Visualization:** Ardashir Mohammadzadeh.

**Writing – original draft:** Amin Taghieh, Ardashir Mohammadzadeh.

**Writing – review & editing:** Afef Fekih.

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
