## [Decision Letter · Decision Letter 0]

22 Nov 2021

PONE-D-21-35036H∞-based control of multi-agent systems: Time-delayed signals, unknown leader states and switching graph topologiesPLOS ONE

Dear Dr. Mohammadzadeh,

Thank you for submitting your manuscript to PLOS ONE. After careful consideration, we feel that it has merit but does not fully meet PLOS ONE’s publication criteria as it currently stands. Therefore, we invite you to submit a revised version of the manuscript that addresses the points raised during the review process.

We look forward to receiving your revised manuscript.

Kind regards,

Fei Chen

Academic Editor

PLOS ONE

Journal Requirements:

2. Please ensure that you refer to Figure 3 and Figure 4 in your text as, if accepted, production will need this reference to link the reader to the figure.

Additional Editor Comments:

Two reviews have been collected. Both reviewers agree that the paper contains publishable results, but they also raised critical comments. Some of them are summarized below.

1) Literature review is not sufficient.

2) The motivation of this work is unclear. For instance, why local stability is studied instead of global stability?

3) Practical examples need to be presented.

4) Math symbols are not defined.

5) Discussions/clarifications of the results need to be given.

Reviewers' comments:

Reviewer's Responses to Questions

**Comments to the Author**

1. Is the manuscript technically sound, and do the data support the conclusions?

Reviewer #1: Partly

Reviewer #2: Yes

2. Has the statistical analysis been performed appropriately and rigorously? 

Reviewer #1: No

Reviewer #2: Yes

3. Have the authors made all data underlying the findings in their manuscript fully available?

Reviewer #1: Yes

Reviewer #2: Yes

4. Is the manuscript presented in an intelligible fashion and written in standard English?

Reviewer #1: Yes

Reviewer #2: Yes

5. Review Comments to the Author

Reviewer #1: The authors in this paper developed H∞-based leader-following scheme for multi-agent systems.

However, I have some remarks that if they are answered, it can improve the manuscript:

In the section “Simulations’ authors chosen some parameters for simulation, but absent any information of justification of chosen parameters. Why the time-delay was selected as 200(ms)? The parameters must be related with any physical model.

Must be added expended discussion and comments about figures 5-15.

In conclusion authors write about stability of the suggested distributed observer and controller, but in the paper are absent any estimation of criteria stability.

My best regards,

Reviewer #2: 1. The abstract and conclusion can be enhanced by capturing the main results as well as the significance.

2. Survey of existing literature is not sufficient. It would useful to include in the Introduction of the paper some

discussion on other possible real applications of the obtained results.

3. The authors have to state the weakness in previous works which motivated this study.

4. Some abbreviations need to be extended when mentioned for the first time, like LMASs, NMASs, etc.

5. The local asymptotic stability is goal of this study. Why the authors did not address the global asymptotic stability.

6. The design analysis is very good, but the choice of design parameters is still a challenging problem. I suggest to

perform one of the modern techniques in tuning these design parameters.

7. The author needs to give some practical examples to support the work.

8. This study has to conduct a comparison in performance with other strategies.

9. Many symbols of equations are not defined. See Eq.(6)-Eq.(11).

10. The switching from Eq.(20) to Eq.(21) is not clear.

11. The authors have to explain the derivation of Eq.(22).

12. The authors has to explain how the alpha's and beta's parameters have been chosen.

13. The discussion of results needs some clarification.

14. The conclusion has to be extended with future work.

6. PLOS authors have the option to publish the peer review history of their article (what does this mean?). If published, this will include your full peer review and any attached files.

Reviewer #1: No

Reviewer #2: **Yes: **AYAD AL-DUJAILI

---

## [Author Response · Author response to Decision Letter 0]

15 Dec 2021

Please see the responses in the end of revised manuscript

---

## [Decision Letter · Decision Letter 1]

11 Jan 2022

H∞-based control of multi-agent systems: Time-delayed signals, unknown leader states and switching graph topologies

PONE-D-21-35036R1

Dear Dr. Mohammadzadeh,

We’re pleased to inform you that your manuscript has been judged scientifically suitable for publication and will be formally accepted for publication once it meets all outstanding technical requirements.

Kind regards,

Fei Chen

Academic Editor

PLOS ONE

Additional Editor Comments (optional):

The reviewers have no further comments. I am pleased to recommend the paper for publication.

Reviewers' comments:

Reviewer's Responses to Questions

**Comments to the Author**

1. If the authors have adequately addressed your comments raised in a previous round of review and you feel that this manuscript is now acceptable for publication, you may indicate that here to bypass the “Comments to the Author” section, enter your conflict of interest statement in the “Confidential to Editor” section, and submit your "Accept" recommendation.

Reviewer #1: All comments have been addressed

Reviewer #2: All comments have been addressed

2. Is the manuscript technically sound, and do the data support the conclusions?

Reviewer #1: Yes

Reviewer #2: Yes

3. Has the statistical analysis been performed appropriately and rigorously? 

Reviewer #1: Yes

Reviewer #2: Yes

4. Have the authors made all data underlying the findings in their manuscript fully available?

Reviewer #1: Yes

Reviewer #2: Yes

5. Is the manuscript presented in an intelligible fashion and written in standard English?

Reviewer #1: (No Response)

Reviewer #2: Yes

6. Review Comments to the Author

Reviewer #1: (No Response)

Reviewer #2: I have read the article thoroughly and the authors have improved the manuscript based on previous comments, and I have no more. Therefore, the current version of the paper is qualified to be accepted in the Journal.

7. PLOS authors have the option to publish the peer review history of their article (what does this mean?). If published, this will include your full peer review and any attached files.

Reviewer #1: No

Reviewer #2: No

---

## [Editor Report · Acceptance letter]

21 Feb 2022

PONE-D-21-35036R1 

H∞-based control of multi-agent systems: Time-delayed signals, unknown leader states and switching graph topologies 

Dear Dr. Mohammadzadeh:

I'm pleased to inform you that your manuscript has been deemed suitable for publication in PLOS ONE. Congratulations! Your manuscript is now with our production department. 

Kind regards, 

on behalf of

Dr. Fei Chen 

Academic Editor

PLOS ONE